# Genome-Wide Identification of LOX Gene Family and Its Expression Analysis under Abiotic Stress in Potato (*Solanum tuberosum* L.)

**DOI:** 10.3390/ijms25063487

**Published:** 2024-03-20

**Authors:** Jinyong Zhu, Limin Chen, Zhitao Li, Weilu Wang, Zheying Qi, Yuanming Li, Yuhui Liu, Zhen Liu

**Affiliations:** 1State Key Laboratory of Aridland Crop Science, Gansu Agricultural University, Lanzhou 730070, China; zhujy_salvare@tom.com (J.Z.); chenlmgsau@foxmail.com (L.C.); lizt1225@163.com (Z.L.); wangwlcn@foxmail.com (W.W.); qzygsau@foxmail.com (Z.Q.); 2College of Agronomy, Gansu Agricultural University, Lanzhou 730070, China; 3College of Horticulture, Gansu Agricultural University, Lanzhou 730070, China; liyuanm@gsau.edu.cn

**Keywords:** abiotic stress, gene expression, LOX gene family, phylogenetic analysis, potato

## Abstract

The lipoxygenases (LOXs) are non-heme iron-containing dioxygenases that play an important role in plant growth and defense responses. There is scarce knowledge regarding the LOX gene family members and their involvement in biotic and abiotic stresses in potato. In this study, a total of 17 gene family members (StLOXs) in potato were identified and clustered into three subfamilies: 9-LOX type I, 13-LOX type I, and 13-LOX type II, with eleven, one, and five members in each subfamily based on phylogenetic analysis. By exploiting the RNA-seq data in the Potato Genome Sequencing Consortium (PGSC) database, the tissue-specific expressed and stress-responsive *StLOX* genes in double-monoploid (DM) potato were obtained. Furthermore, six candidate *StLOX* genes that might participate in drought and salt response were determined via qPCR analysis in tetraploid potato cultivars under NaCl and PEG treatment. Finally, the involvement in salt stress response of two *StLOX* genes, which were significantly up-regulated in both DM and tetraploid potato under NaCl and PEG treatment, was confirmed via heterologous expression in yeast under salt treatment. Our comprehensive analysis of the StLOX family provides a theoretical basis for the potential biological functions of StLOXs in the adaptation mechanisms of potato to stress conditions.

## 1. Introduction

Lipoxygenase (LOX) is a dioxygenase containing non-heme iron [1], which can catalyze the oxygenation of unsaturated fatty acids and their corresponding esters in plants. The resultant product of the oxygenation reaction catalyzed by LOXs is hydroperoxides. The secondary metabolites of these hydroperoxides, such as jasmonic acid (JA), green leafy volatiles (GLVs), and divinyl ether, play important roles in regulating seed germination [2], root growth [3], fruit ripening [4], pathogen infection [5], pest resistance [6], and response to abiotic stress [7].

Lipoxygenase is ubiquitously present in plants, animals, algae, and fungi [8,9,10,11]. Plants harbor LOX proteins that possess two conserved structural domains: the PLAT/LH2 domain located at the N-terminus and the lipoxygenase domain situated at the C-terminus [12]. The catalytic site of the enzyme resides within the lipoxygenase structural domain, which encompasses 38 amino acids [His-(X)4-His-(X)4-His-(X)17-His-(X)8-His], including 5 highly conserved histidine (His) residues [13]. The LOX family can be classified into 9-LOX and 13-LOX according to the reaction position of carbon atoms when they catalyze the oxidation of substrates. In addition, the classification of the LOX family into Type I and Type II is determined by two factors: the degree of similarity in amino acid sequences and whether a chloroplast transit peptide is present at the N-terminal. Type I LOX sequences exhibit high similarity and are devoid of transit peptides, whereas Type II LOX sequences display moderate similarity and possess chloroplast transit peptides at the N-terminus [1,14].

LOX enzyme plays a pivotal role in oxidizing unsaturated fatty acids, thereby exerting a crucial influence on secondary metabolism and vital plant processes. Six LOX members have been found in *Arabidopsis thaliana*, of which AtLOX2, AtLOX3, AtLOX4, and AtLOX6 belong to 13-LOX and AtLOX1 and AtLOX5 belong to 9-LOX [15]. The four 13-LOX enzymes are involved in jasmonate synthesis after leaf injury [16], and the 9-LOX enzymes may control the emergence of lateral roots by producing 9-Hydroxyoctadecatrienoic acid (9-HOT), thus regulating root development [17]. Hou et al. [18] found that the overexpression of the persimmon (*Diospyros kaki* Thunb.) gene *DkLOX3* increased the resistance of transgenic *Arabidopsis* to *Botrytis cinerea*. The overexpression of pepper *CaLOX1* could rapidly remove ROS and induce the expression of ABA biosynthesis-related genes, which improved the tolerance of transgenic *Arabidopsis* to drought and salt stress [19]. In tomato (*Solanum lycopersicum*), the overexpression of *TomloxD* (*Solyc03g122340*, 13-LOX) exhibited enhanced resistance against the *Cladosporium fulvum* and high-temperature stress, which was due to an increase in lipoxygenase activity, endogenous levels of JA, and the expression of defense genes such as *LeHSP90*, *LePR1*, *LePR6*, and *LeZAT* [20,21]. Numerous LOXs have been shown to play a pivotal role in plants’ ability to respond to various stressors, encompassing both biotic stressors like pathogens [20] and aphids [22] and abiotic stressors such as cold [23], heat [24], salt, and drought [19]. Currently, surprisingly little research has focused on elucidating the LOX gene family in potato, despite the extensive identification and analysis of LOXs in diverse plants, including 8 members in pepper (*Capsicum annuum*) [25], 11 members in radish (*Raphanus sativus*) [26], 20 members in poplar (*Populus trichocarpa*) [27], and 11 members in tea tree (*Camellia sinensis*) [23].

Potato, which originates from the Andean region of Bolivia and Peru, is the third most-produced food crop, following rice (*Oryza sativa*) and wheat (*Triticum aestivum*) [28]. Potato is widely cultivated worldwide because of its tolerance to barrenness, wide adaptability, and high and stable yield. Potato tuber production is impacted by various biotic and abiotic stressors, including pests and diseases, high temperature, salt stress, and drought stress, all of which significantly hinder the growth and progress of the potato industry.

In this study, 17 members of the LOX gene family were identified through comprehensive genome-wide analysis of potato. Then, the *StLOX* gene structure and duplication events, phylogenetic relationships, and conserved motifs of proteins were systematically investigated. Moreover, the expression of *StLOX* genes in different tissues of double-monoploid (DM) potato, as well as under different stresses and hormone treatments, were analyzed based on RNA-seq data in the PGSC database. Furthermore, some candidate genes in tetraploid potato cultivars that responded to NaCl and PEG were confirmed via qPCR. Of these, two candidate genes, *Soltu.DM.01G002140* and *Soltu.DM.01G038840*, that might be involved in the response to salt stress were verified via the yeast heterologous expression method.

## 2. Results

### 2.1. Identification of StLOXs

By employing HMM 3.1, we conducted a search of the potato genome sequence to identify proteins containing the lipoxygenase domain (PF00305). We utilized the SMART and NCBI Batch CD-search databases to screen for sequences that harbor the two conserved domains of PLAT/LH2 and lipoxygenase. At the end of the analysis, a complete set of 17 members belonging to the StLOX gene family was identified. They were unevenly distributed in 6 out of 12 chromosomes. Chromosome 1 harbored the highest count of genes, encompassing eight *StLOX* genes, while chromosomes 3, 5, and 12 exhibited the lowest representation, with only a single *StLOX* gene for each chromosome (Figure 1). The physical and chemical property analyses of StLOX members showed that the lengths of StLOX proteins ranged from 794 amino acids (Soltu.DM.08G005470) to 914 amino acids (Soltu.DM.03G037120), the molecular weights ranged between 89.90 kDa (Soltu.DM.08G005470) and 103.59 kDa (Soltu.DM.03G037120), and the pI values ranged from 5.15 (Soltu.DM.01G038910) to 8.6 (Soltu.DM.05G011130) (Appendix A).

### 2.2. Phylogenetic Analysis and Classification of StLOXs

To understand the evolutionary relationship of the StLOX gene family, 60 LOX proteins from 6 species (potato, *Arabidopsis*, tomato, cabbage, rice, and maize) were compared, and a phylogenetic tree was constructed. The results showed that these 60 LOX proteins were classified into 3 subfamilies: 9-LOX type I, 13-LOX type I, and 13-LOX type II, in which there were 11, 1, and 5 members of the StLOX family, respectively. The three subfamilies were further divided to five subgroups: C1-C5; among them, C1 and C2 belonged to 13-LOX type II, C3 and C4 belonged to 13-LOX type I, and C5 belonged to 9-LOX subfamily. Seventeen StLOXs were characterized into four subgroups, except for the C4 subgroup (Figure 2). Most of the StLOX members were closely related to those of tomatoes, which all belonged to the Solanaceae family, but relatively distant from those of other dicotyledonous and monocotyledonous species.

### 2.3. Gene Structure and Conserved Motifs of StLOXs

To gain insight into the architecture of the *StLOX* genes, the introns and exons of its family members were subjected to analysis and visualization (Figure 3B). Of the 17 *StLOX* genes, only 1 member (*Soltu.DM.03G037120*) contained 7 introns, 12 members contained 8 introns, and 4 members (*Soltu.DM.01G002150*, *Soltu.DM.01G002140*, *Soltu.DM.12G02875*, and *Soltu.DM.08G005470*) contained 9 introns. Additional analysis of the conserved motifs present in the StLOXs was carried out using MEME. The findings revealed the identification of ten distinct motifs, designated as Motif 1–Motif 10 (Figure 3C and Appendix A). In total, 10 9-LOX members contained all 10 motifs except for Soltu.DM.01G038930, while all 6 13-LOX members lacked motif 6. The conserved motifs of potato LOX family proteins are highly conserved, as evidenced by the identical arrangement order among StLOX members. Motif 1 comprises 50 amino acids residues and contains the LOX domain [His-(X)4-His-(X)4-His-(X)17-His-(X)8-His], which binds to ferric ions to form the active centre of the enzyme. 

### 2.4. Gene Duplication Events in the StLOX and Analysis of Synteny with Other Species

The duplication of genes plays a pivotal role in the enlargement of gene families and the diversification of their functions. During the current investigation, we identified four pairs of *StLOX* genes (6/17, 35.29%) as being tandemly duplicated. Specifically, chromosome 1 harbored three such pairs, while chromosome 8 contained one such pair (Figure 1). 

The assessment of selection pressure on genes was conducted by calculating the ratio between non-synonymous mutations (Ka) and synonymous mutations (Ks). A Ka/Ks ratio equal to 1 signified neutral evolution, while a ratio less than 1 signified the influence of purification selection on the gene. Conversely, a Ka/Ks ratio exceeding 1 hinted at the presence of positive selection. For the tandemly duplicated genes, the Ka/Ks values ranged from 0.1558 to 0.3906, with a mean of 0.3078 (Appendix A). These values, all being below 1, provide further evidence that the evolution of these genes was significantly influenced by purifying selection.

To elucidate the potential evolution of the LOX gene family in tomato, *Arabidopsis*, cabbage, rice, maize, and potato, we conducted a detailed analysis of their orthologous genes (Figure 4). Consequently, we identified 11, 3, 3, 0, and 1 pairs of orthologs, respectively. The corresponding Ka/Ks values ranged from 0.0554 to 0.4125 (Figure 4 and Appendix A). Notably, all these ratios were below 1, indicating that these genes underwent evolutionary processes under purifying selection. 

### 2.5. Patterns of StLOX Gene Expression across Different Tissues in DM Potato

Leveraging RNA-seq data retrieved from the PGSC database, a study investigated the unique expression patterns of the *StLOX* gene across diverse tissues of DM potato, ranging from sepals and leaves to roots and shoots, as well as callus, stolons, mature flowers, tubers, petioles, petals, stamens, carpels, the insides of fruits, and both mature and immature fruits (Figure 5 and Appendix A). Three *StLOXs* (*SoltuDM.03G037120*, *Soltu.DM.09G024180*, *Soltu.DM.05G011130*) had high expression levels in all tissues (FPKM > 3). Certain *StLOX* genes had a tissue-specific expression, such as *Soltu01G038900* and *Soltu01G038910*, which were highly expressed in fruit, while three *StLOX* genes (*Soltu08G010990*, *Soltu08G005480*, and *Soltu03G0317120*) were highly expressed in stolons and three *StLOX* genes (*Soltu08G005480*, *Soltu08G005440*, and *Soltu08G005470*) were highly expressed in tubers.

### 2.6. Patterns of StLOX Gene Expression in Response to Stress Conditions and Hormonal Treatments

Using transcriptome data downloaded from PGSC, the expression profiles of *StLOX* genes in DM potato under various treatments, including salt, mannitol, heat, *P. infestans*, *β*-aminobutyric acid (BABA), and benzo-2,1,3-thiadiazole (BTH), were analyzed. The analysis revealed that three, two, and five genes exhibited differential expression (FPKM > 1, |log_2_FC| > 1) under salt, mannitol, and heat treatments, respectively. Among them, three genes (*Soltu.DM.01G038930*, *Soltu.DM.08G005480*, and *Soltu.DM.08G010990*) were differentially expressed under two stress treatments, both belonging to the 9-LOX subfamily. Four genes (*Soltu.DM.01G002140*, *Soltu.DM.01G038910*, *Soltu.DM.03G037120*, and *Soltu.DM.01G038840*) showed differential expression under a single abiotic stress. The expression of nine *StLOX* genes was significantly altered under biotic stress conditions (FPKM > 1, |log_2_FC| > 1). Eight *StLOX* genes (*Soltu.DM.01G038930*, *Soltu.DM.08G005480*, *Soltu.DM.01G002140*, *Soltu.DM.03G037120*, *Soltu.DM.12G028750*, *Soltu.DM.09G024180*, *Soltu.DM.05G011130*, and *Soltu.DM.08G010990*) responded exclusively to one specific stress. Only one *StLOX* gene (*Soltu.DM.01G002150*) exhibited differential expression under two different stresses (Figure 6 and Appendix A).

Moreover, RNA-seq data retrieved from PGSC were utilized to investigate the transcriptional patterns of StLOX genes in response to hormone treatments, encompassing ABA, IAA, GA3, and BAP. The results showed that four, one, and two *StLOX* genes responded to ABA, GA3, and BAP treatments (FPKM > 1, |log_2_FC| > 1), respectively (Figure 6 and Appendix A). Two *StLOX* genes responded to two or more hormone treatments. The expression of *Soltu.DM.03G037120* was up-regulated in response to ABA and GA3 treatments, while it was down-regulated under BAP treatment. Similarly, the expression of *Soltu.DM.08G010990* was up-regulated by ABA and down-regulated by BAP. Additionally, the expressions of *Soltu.DM.01G038840* and *Soltu.DM.09G024180* were elevated upon ABA treatment. 

### 2.7. Expression of Six StLOX Genes in Potato Plantlets under NaCl and PEG Treatments

To validate the responsiveness of six *StLOX* genes to abiotic stresses in tetraploid potato, we conducted a qPCR analysis to assess their expression patterns in potato plantlets (CIP98 clone) exposed to NaCl and PEG treatments, given that these genes exhibit differential expression in DM potato under abiotic stresses. qPCR analysis showed that four *StLOX* genes (*Soltu.DM.08G005480*, *Soltu.DM.01G002140*, *Soltu.DM.03G037120*, and *Soltu.DM.01G038840*) were significantly up-regulated compared with CK under NaCl treatment. Of them, *Soltu03G037120* was the most significantly up-regulated gene, showing a 2.67 times higher expression level than that of the control. PEG treatment led to the up-regulation of these four genes, with the most pronounced increase observed in *Soltu.DM.01G038840*, which exhibited a 2.77-fold higher expression compared to the control (Figure 7 and Appendix A). The results further confirmed that these three *StLOX* genes were involved in the responses to abiotic stress, and their functions are worthy of further study.

### 2.8. Cloning and Functional Analysis of StLOX Gene

To further verify whether the three candidate *StLOX* genes (*Soltu.DM.01G002140*, *Soltu.DM.03G037120* and *Soltu.DM.01G038840*) were involved in abiotic stresses, the CDS fragments of *Soltu.DM.01G002140* (2700bp), *Soltu.DM.03G037120* (2745bp), and *Soltu.DM.01G038840* (2589bp) were cloned by using the cDNA of potato clone “CIP98” as a template, then ligated into the yeast expression vector pYES2-NTB and treated with NaCl of various concentrations for 7 days. The yeast pYES2 was used as a control. Under normal conditions, the growth statuses of three LOX-overexpressing transgenic yeasts were indistinguishable from that of pYES2 yeast cells. The results indicated that there were no differences in the growth statuses of three LOX-overexpressing transgenic yeast or pYES2 yeast cells under normal conditions. However, under 0.5 M and 1.0 M NaCl treatments, the yeast cells overexpressing *Soltu.DM.01G002140* exhibited optimal growth activity, followed by the yeast cells overexpressing *Soltu.DM.01G038840*. When under 1.5 M NaCl stress, the yeast cells overexpressing *Soltu.DM.01G002140* could still grow properly even when diluted 10^3^ and 10^4^ times (Figure 8). The findings demonstrated that *Soltu.DM.01G002140* and *Soltu.DM.01G038840* genes enhance salt stress tolerance in transgenic yeast cells, further confirming their involvement in abiotic stress responses.

## 3. Discussion

Numerous studies have been carried out on the LOX gene families in various plant species, revealing their possible functions in promoting plant growth, development, and defensive responses. The present study identified and analyzed 17 StLOXs in terms of phylogenetic relationships, gene and protein structures and expression profiles. The results showed that StLOX genes play a subtle tissue-specific role during development and are essential in plant stress response. Thus, the data presented herein will provide a useful basis for the functional characterization of *StLOX* genes.

The primary forces driving the emergence of new family members and novel functions in plant evolution are undergoing tandem duplication and segmental duplication [29]. A total of four pairs of tandemly duplicated genes were identified in potato, all of which belonged to the 9-LOX type I. This implied that the evolution of *StLOX* genes was dominated by tandem replication, which plays an especially important role in the expansion of the 9-LOX gene family. Additionally, analysis of multi-species collinearity revealed that certain *LOX* genes are highly conserved during the evolution of plants. For example, *Soltu.DM.03G037120* is homologous with tomato *Solyc03g122340*, *Arabidopsis At1G17420*, cabbage *Bo8g104870*, and corn *Zm00001d027893*; *Soltu.DM.09G024180* is homologous with tomato *Solyc09g075860*, *Arabidopsis At3G22400*, and cabbage *Bo1g105160*. These *LOX* genes may perform similar functions in different plants.

LOX proteins play a crucial role in plant growth, development, and resilience against adverse environmental conditions [30,31,32]. *Solt.DM.08G005440* and *Soltu.DM.08G005470* are specifically expressed in tubers but not in stolons, and they may be involved in the development processes of tubers. In response to stress, three 9-LOX genes (*Soltu.DM.08G010990*, *Soltu.DM.01G038930*, and *Soltu.DM.08G005480*) exhibited differential expression patterns, analogous to those observed in radish upon exposure to NaCl and PEG stress [26]. Through their study, Hu et al. revealed that overexpressing *TomloxD* (*Solyc03g122340*) in tomato plants significantly boosts the production of endogenous Jasmonic acid, subsequently improving the plant’s tolerance of both biotic and abiotic stress [20]. As a homologous gene of *Solyc03g122340*, *Soltu.DM.03G037120* showed differential expression under the treatment of mannitol, multiple hormones, and BTH, which may have potential applications in crop resistance to biotic and abiotic stress. ZS-3, an eco-friendly salt tolerance inductor, augments the growth of plants under salt stress. *AtLOX2* is induced by ZS-3 and enhances the salt tolerance capacity of plants through the jasmonic acid pathway [33]. We found that Soltu.DM.01G002140 was in the same subgroup as AtLOX2, which was differentially up-regulated under NaCl and ABA treatment and differentially down-regulated under heat stress and BABA treatment. The overexpression of *Soltu.DM.01G002140* significantly improved the salt (NaCl) tolerance of yeast (INVsc1). Despite their role in catalyzing the oxygenation of polyunsaturated fatty acids in vivo, it remains to be experimentally confirmed in future investigations whether LOXs are activated by distinct external signals to maintain the metabolite balance in potato.

## 4. Materials and Methods

### 4.1. Identification of LOX Gene Family Members and Analysis of Protein Physicochemical Properties in Potato

The potato genome data (PGSC_DM_v6.1) were downloaded from the PGSC (Potato Genome Sequencing Consortium, http://spuddb.uga.edu/ accessed on 15 September 2023) database. Genome resources for *Arabidopsis thaliana*, *Solanum lycopersicum*, *Brassica oleracea*, *Oryza sativa*, and *Zea mays* were downloaded from the Ensembl Plants (http://plants.ensembl.org/index.html accessed on 18 September 2023) database. The protein sequences of six AtLOXs were downloaded from TAIR (https://www.arabidopsis.org/ accessed on 20 September 2023) [15]. HMM profiles for the LOX domain (PF00305) were downloaded from the InterPro [34] website (https://www.ebi.ac.uk/interpro/entry/pfam/PF00305/ accessed on 25 September 2023). HMMER 3.1 software (http://hmmer.org/download.html accessed on 25 September 2023) was then employed to search for the LOX domain in the proteins of each species. All candidate sequences underwent artificial selection through SMART [35] (http://smart.embl-heidelberg.de/ accessed on 26 September 2023) and NCBI Conserved Domain Data (CDD) [36] (https://www.ncbi.nlm.nih.gov/Structure/cdd/wrpsb.cgi accessed on 26 September 2023), and sequences that did not contain LOX and PLAT/LH2 domains (PF01477) were excluded. The ExPasy website [37] was used to predict the molecular weight, theoretical isoelectric point, and amino acid number of the StLOX proteins. Mapchart software [38] (version 2.32, http://www.wageningenur.nl/en/show/Mapchart.htm accessed on 15 October 2023) was utilized to generate a map depicting the localization of *StLOX* genes on potato chromosomes.

### 4.2. Phylogenetic Tree Construction of LOXs

A phylogenetic tree encompassing 17 StLOXs, 6 AtLOXs, 10 OsLOXs, 8 ZmLOXs, 9 BoLOXs, and 10 SlLOXs was constructed by employing MEGA7.0 software using the maximum likelihood method. The Poisson model was utilized, and 1000 bootstrap repetitions were conducted [39]. In the meantime, a phylogenetic tree encompassing 17 members of StLOX was constructed, employing an identical methodology.

### 4.3. Conserved Motif and Gene Structural Characterization of StLOXs

The motifs present in StLOX proteins were analyzed via the MEME Version 5.5.5 program (https://meme-suite.org/meme/tools/meme accessed on 20 October 2023). A maximum of 10 motifs were allowed, with lengths varying from 6 to 50 amino acids [40]. Furthermore, the GSDS 2.0 (http://gsds.gao-lab.org accessed on 21 October 2023) tool was used to offer a graphical depiction of the gene structures of the *StLOXs* [41].

### 4.4. Duplication Events and Syntenic Patterns of LOX Genes

The MCScanX v1.5.1 [42] was used to identify *StLOX* duplication and collinearity analysis, and Circos v0.69 [43] was applied to provide a visual representation of the syntenic relationships between LOX genes in various species, including *Solanum tuberosum*, *Arabidopsis thaliana*, *Solanum lycopersicum*, *Brassica oleracea*, *Oryza sativa*, and *Zea mays.* Following that, the KaKs Calculator 2.0 [44] was employed to determine the non-synonymous (Ka) and synonymous (Ks) substitution rates for each duplicated pair of LOX genes. 

### 4.5. Transcriptional Profile Analysis of StLOXs in Potato

Utilizing published RNA sequencing data sourced from Illumina RNA-seq experiments conducted by PGSC (http://spuddb.uga.edu/ accessed on 15 September 2023) [45], we conducted a comprehensive analysis of the expression patterns of the *StLOX* gene family across various tissues of DM potato (sepals, leaves, roots, shoots, callus, stolons, mature flowers, tubers, petioles, petals, stamens, carpels, mature fruits, immature fruits, and the insides of fruits). Additionally, we explored the expression patterns in DM potato under stress conditions (salt treatment: 150 mM NaCl, 24 h; mannitol treatment: 260 µM mannitol, 24 h; heat treatment: 35 °C, 24 h; *P. infestans*-infected leaves, 24/48/72 h; *β*-aminobutyric acid-treated leaves, 24/48/72 h; Benzo-2,1,3-thiadiazole-treated leaves, 24/48/72 h) and during hormone treatments (abscisic acid treatment: 50 µM, 24 h; indole acetic acid treatment: 10µM, 24 h; gibberellin treatment: 50 µM, 24 h; 6-benzylamino purine treatment: 10 µM, 24 h). TBtools v2.030 software [46] was used to draw the heatmap. 

### 4.6. Plant Materials and Treatments

The experimental material comprised a tetraploid potato plantlet named “CIP397098.12” (CIP98), which was obtained from the International Potato Center (CIP). The tissue cultured seedlings were placed in Murashige and Skoog (MS) liquid media containing 3% sucrose (*w*/*v*, pH 6.0 ± 0.5) and incubated in a light incubator for three weeks (23 ± 1 °C, 12 h light/12 h dark photoperiod). For the simulated drought and salinity stress treatments, these plantlets were treated with 10% PEG 6000 and 100 mM NaCl for 24 h, respectively. Each triangular flask contained five plantlets, and each of the three replicates contain three triangular flasks. The sample was immediately frozen in liquid nitrogen and stored at −80 °C for RNA extraction.

### 4.7. Extraction of RNA and qPCR

RNA extraction was carried out using the RNA extraction kit (TaKaRa MiniBEST Plant RNA Extraction Kit, Code No. 9769, TaKaRa, Kusatsu, Japan) from all samples, and the integrity and concentration of the RNA were subsequently assessed via agarose gel electrophoresis and Nanodrop ND-2000 spectrophotometer (Nanodrop Technologies, Wilmington, DE, USA) measurements, respectively. The FastKing gDNA Dispelling RT SuperMix (Tiangen KR118, Beijing, China) was used to synthesize the first-strand cDNA, with the RNA serving as the template. The qPCR experiments were conducted on the Bio-Rad CFX96 (Bio-Rad, Hercules, CA, USA), employing the SuperReal PreMix Plus kit (SYBRGreen FP217, Tiangen, Beijing, China). Each assay was carried out with three technical replicates. The PCR program was as follows: 95 °C for 2 min, followed by 40 cycles of 95 °C for 5 s and 60 °C for 15 s and dissociation curve detection under 65–95 °C. Additionally, the internal control gene *StEF-1α* (AB061263) was employed [47]. The quantification of *LOX* gene expression was determined via the 2^−∆∆Ct^ method [48]. The primers are shown in Appendix A.

### 4.8. Cloning of the 3 StLOX Genes

The CDS sequences of the candidate genes *Soltu.DM.01G002140*, *Soltu.DM.03G037120,* and *Soltu.DM.01G038840* were amplified by using “CIP98” cDNA as a template. The primers were listed in Appendix A, and the amplified sequences are shown in Appendix A. The gene amplification process was carried out in a 50 μL reaction mixture by utilizing PrimeSTAR Max DNA Polymerase (TaKaRa R045Q, Japan). The linearization of the pYES2 vector was performed using *Eco*RI and *Bam*HI (TaKaRa 1040S/1010S, Japan), and the target gene was cloned into the pYES2 vector by using In-Fusion Snap Assembly cloning kits (TaKaRa 638947, Japan).

### 4.9. Salt Stress Tolerance Verification of Candidate Genes in Yeast

Yeast has been proven to be an effective system for evaluating gene tolerance [49]. Three candidate genes, *Soltu.DM.01G002140*, *Soltu.DM.03G037120,* and *Soltu.DM.01G038840*, in the pYES2 vector were transformed into yeast strain INVsc1 and coated on the corresponding yeast medium (SD-Ura). A single colony was picked up after incubation at 30 °C for 3 days, and positive colonies were identified using PCR. For the salt stress assay, transformed yeast cells were incubated in SG/-Ura liquid medium at 30 °C for 16 h (OD_600_ ≈ 0.6); then, these yeast cells were serially diluted 10-fold (10^0^, 10^−1^, 10^−2^, 10^−3^, and 10^−4^) with sterile water and dripped in the SG/-Ura medium treated with different concentrations of NaCl (0 M, 0.5 M, 1.0 M, 1.5 M). 

## 5. Conclusions

This study provides a thorough analysis of the StLOX gene family, encompassing its phylogeny, gene structure, motif arrangement, evolution, and patterns of gene expression under various abiotic and biotic stresses. This study found that the distribution of StLOX members on the six chromosomes was uneven. The primary cause of the enlargement of the StLOX gene family was tandem duplication occurrences. Furthermore, qPCR analysis of StLOX expression in DM and tetraploid potato under abiotic stresses indicated that three *StLOX* genes were involved in responses to salt and drought stresses, and the overexpression of *Soltu01G002140* and *Soltu01G038840* enhanced the salt tolerance of yeast cells. These results provide a theoretical basis for further understanding the function of the StLOX gene family in potato.

## Figures and Tables

**Figure 1 ijms-25-03487-f001:**
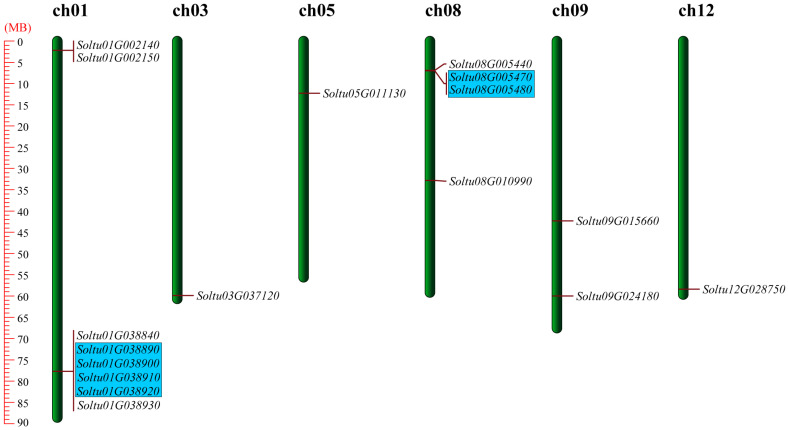
Seventeen *StLOX* genes were distributed in six chromosomes. The blue area marks tandem duplicated genes. The transcript IDs are translated into their respective gene IDs, and ‘Soltu.DM.’ is abbreviated to ‘Soltu’ for brevity’s sake.

**Figure 2 ijms-25-03487-f002:**
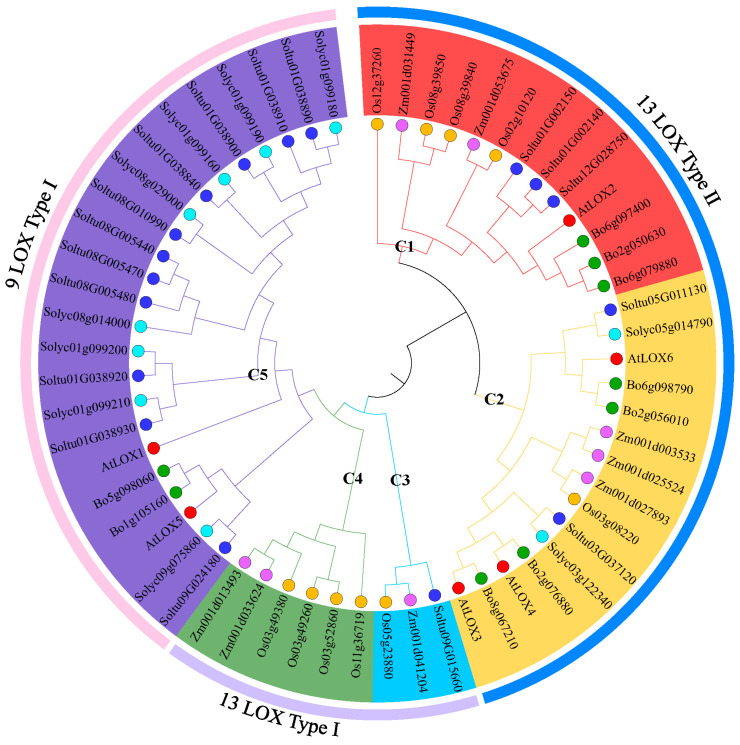
Phylogenetic tree of LOX proteins of six species. Five subgroups (C1–C5) are represented by five distinct colored lines (red, orange, blue, green, and purple). The varying colored circles signify members of different species: red circles represent Arabidopsis, blue circles represent potato, light blue circles represent tomato, green circles represent cabbage, orange circles represent rice, and purple circles represent maize.

**Figure 3 ijms-25-03487-f003:**
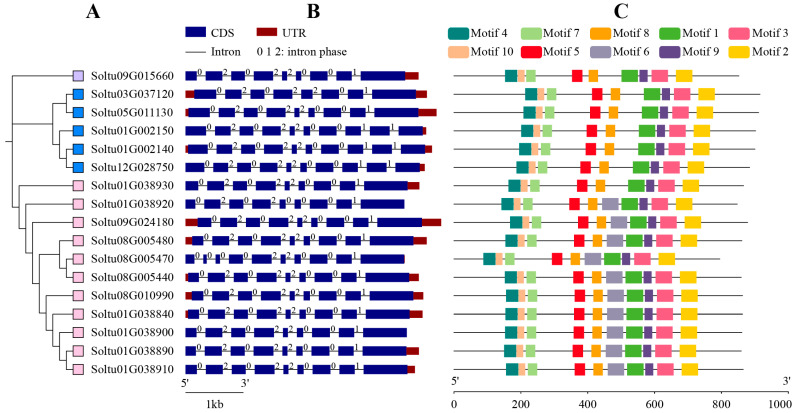
Phylogenetic, gene structure, and motif analysis of the StLOXs. (**A**) Evolutionary tree comprising StLOX proteins. The purple box, blue box, and pink box were used to denote the subfamilies 13-LOX Type I, 13-LOX Type II, and 9-LOX Type I, respectively. (**B**) Exon/intron structure in StLOX genes. Blue boxes represent exons, while black lines represent introns. Similarly, red boxes denote 5′UTR and 3′UTR. The numbers 0, 1, and 2 signify the splicing phases of the introns. (**C**) Conserved motif distribution of StLOX. The 10 different colored boxes represented 10 specific motifs.

**Figure 4 ijms-25-03487-f004:**
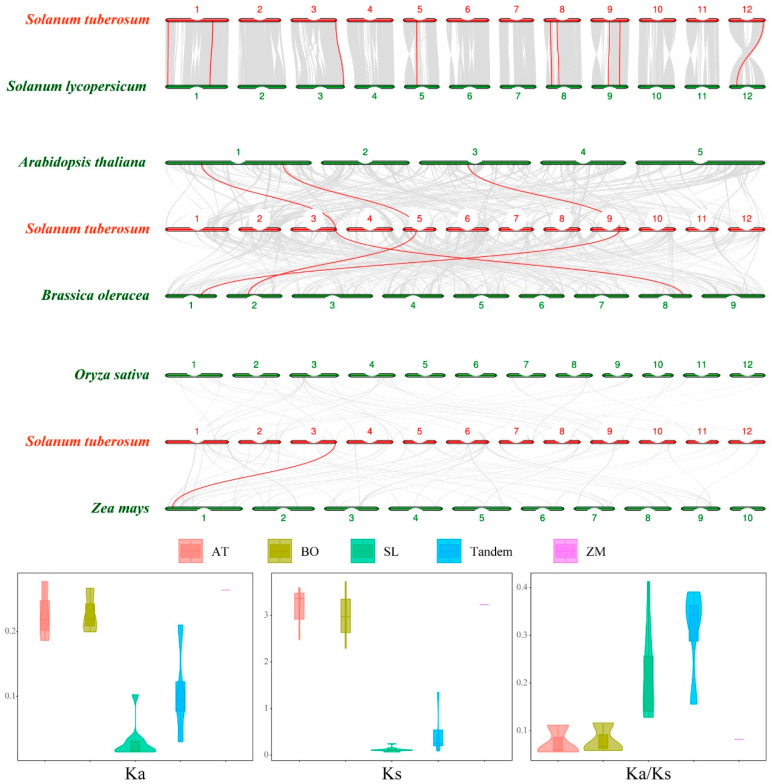
The orthologous relations of the *StLOX* genes among tomato, *Arabidopsis*, cabbage, rice, and maize. A visual representation of the orthology between the *StLOX* gene and multiple plant species, including tomato, *Arabidopsis*, cabbage, rice, and maize, is indicated by a red line. The average values of Ka, Ks, and Ka/Ks for the duplicated genes are presented. The horizontal axis represents tandem duplication in potato (Tandem) and homologous genes between potato and *Arabidopsis* (St-At), cabbage (St-Bo), tomato (St-Sl), and maize (St-Zm), respectively.

**Figure 5 ijms-25-03487-f005:**
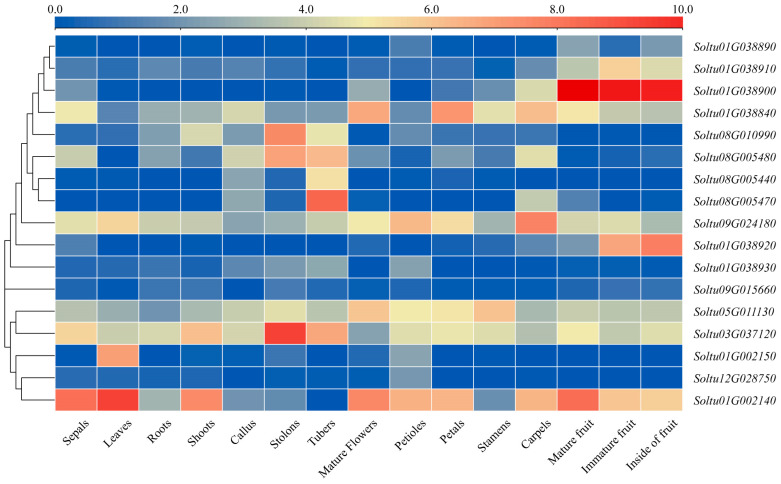
Visual representation of *StLOX* expression across multiple tissues in DM potato. The expression levels of genes are normalized using log_2_FPKM and denoted by colored blocks varying from blue to red.

**Figure 6 ijms-25-03487-f006:**
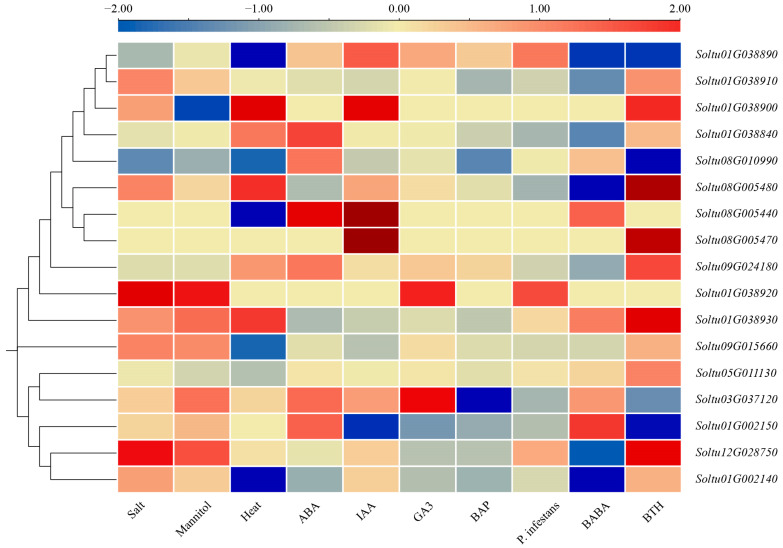
The transcriptional patterns of *StLOX* genes in DM potato, subjected to multiple stresses and treated with hormones including IAA, ABA, GA3, and BAP, are demonstrated. The colored blocks ranging from blue to red indicate the expression levels of these genes, which are normalized using log_2_FC.

**Figure 7 ijms-25-03487-f007:**
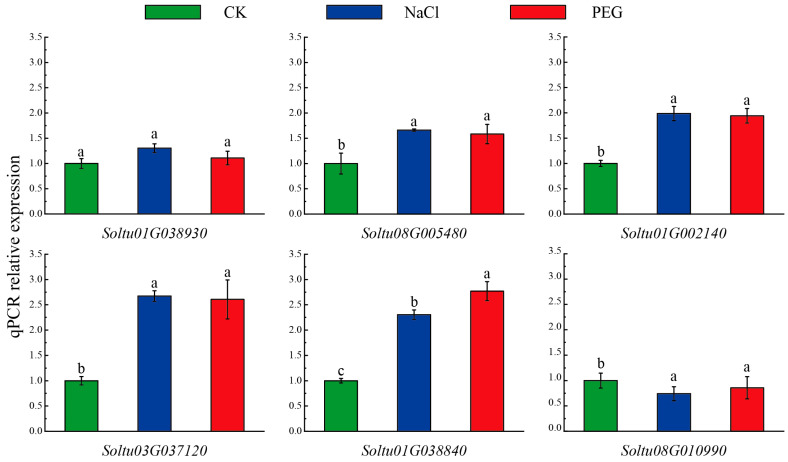
Relative expression levels of *StLOX* genes in potato plantlet leaves under NaCl and PEG treatments for 24 h. Data represent the mean ± standard error of three biological replicates. Error bars represent standard errors. Different letters on the top of the bars indicate significant differences (*p* < 0.05).

**Figure 8 ijms-25-03487-f008:**
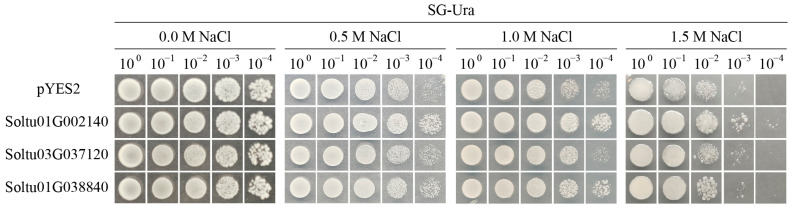
The growth phenotype of three *StLOX*-overexpressing transgenic yeasts under NaCl treatments with concentrations of 0 M, 0.5 M, 1 M, and 1.5 M, respectively.

## Data Availability

Data are contained within the article and Appendix A.

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
