# Peer review of "Genome-Wide Identification of LOX Gene Family and Its Expression Analysis under Abiotic Stress in Potato (Solanum tuberosum L.)"

_ijms, 2024, doi:10.3390/ijms25063487_

Round 1
Reviewer 1 Report
Comments and Suggestions for Authors
Dear Authors,
The reviewed manuscript is in the important subject of analysis of the StLOX gene family, encompassing 407 its phylogeny, gene structure, motif arrangement, evolution, and patterns of gene ex-408 pression under various abiotic and biotic stresses.
In my opinion, the structure of the manuscript is clear. It begins with an Introduction to the issues taken up, in which the Authors point to the studies of Genome-wide identification of LOX gene family and its expression analysis under stress in potato.
The following chapters do not raise any objections - the methodology of the work is described in a very clear way and includes the necessary information on the source of the data used. In the Results chapter, the Authors included an in-depth analysis of the results obtained, which is supplemented by seven figures. The Conclusions chapter contains the most important insights from the analyses.
The Materials and Methods chapter should appear before the results.
Reviewer 2 Report
Comments and Suggestions for Authors
Title: Genome-wide identification of LOX gene family and its expression analysis under abiotic stress in potato (Solanum tu-3 berosum L.)
The current study identified 17 LOXs and conducted phylogenetic analysis of some plants to infer their evolutionary relationships. The findings focused on candidate genes that are involved in salt stress.
The topic is interesting, the experiment is set up well, the writing is good but there are some corrections which are need to be addressed before publication.
The main drawback of the paper is the discussion section. Lines 264-267 are the repeat of literature review. Lines 267-277 are the repeat of conclusion. The rest of the part is not deeply connected to the findings of the paper nor to the finding of other research or the possible involved mechanism. Please revise the section.
Also, provide the key words in alphabetic order and capitalized form.
Line 30 what do you mean “iro”? is it correct? Or it should be iron?
Line 35 after “insect pests” is there any missed word? For example, “resistance”?
Line 72-78 is better to move “Numerous LOXs have been … members in tea tree (Camellia sinensis)[24].” to line 65 after “LePR6, and LeZAT[20, 21].”
Reviewer 3 Report
Comments and Suggestions for Authors
The aim of the work set by the authors was fully achieved. The methodology used is correct, and the study was also, to some extent, extended by comparisons with other plant species. This approach provides a broader range of inferences and allows comparison of the potato plant with related species.
The topic is important and substantively correctly implemented in the experimental part. Identification of the genes responsible for the plant's response to abiotic stresses is of both economic and environmental importance.
"Identification of the genes responsible for the plant's response to abiotic stresses is of both economic and environmental importance." Understanding the level of a plant's response to salinity stress by analyzing the expression of specific genes allows growers to quickly and effectively evaluate breeding material for salinity tolerance. In the context of projected global warming, the increasing risk of longer periods of drought in many regions of the world may lead to increases in salinity levels. This is why breeders of new potato varieties use research results such as those presented in this work. Thanks to the very precise results, breeders will be able to effectively implement this knowledge in breeding practice, which will become a key aspect in developing new potato varieties that are more resistant to low salinity levels.
This research may contribute to understanding and mitigating problems related to stress caused by environmental changes. The conclusion was effectively formulated, indicating a specific practical application of the authors' work results.
Minor comments and suggestions:
Line 91 – "The text continues here." – should be removed.
Figure 7 - please adjust the Y axis range in all charts to a common range (0 - 3.5).
